# Loneliness and cognitive decline among U.S. adults: A stratified analysis of the BRFSS

**Mojisola Fasokun**[1]*, **Temitope Ogundare**[2], **Fadeke Ogunyankin**[3], **Kaelyn Gordon**[4],
**Seun Ikugbayigbe**[5], **Miriam Michael**[6], **Kakra Hughes**[7], **Oluwasegun Akinyemi**[4]

1 Department of Epidemiology, University of Alabama at Birmingham, Birmingham, Alabama, United States of America, 2 Department of Psychiatry, New York Presbyterian Columbia University Medical Center, New York, United States of America, 3 Department of Research Data Science and Analytics, Cook Children's Health Care System: Cook Children's Medical Center, Fort Worth, Texas, United States of America, 4 The Clive O Callender Outcomes Research Center, Howard University College of Medicine, Washington DC, United States of America, 5 Department of Biological Sciences, Eastern Illinois University, Charleston, Illinois, United States of America, 6 Department of Internal Medicine, Howard University College of Medicine, Washington DC, United States of America, 7 Department of Surgery, Howard University College of Medicine, Washington DC, United States of America

* mjfash@uab.edu

## Abstract

### Background

Loneliness is an emerging public health concern linked to adverse mental and physical outcomes. It may play a key role in cognitive aging, yet its population-level association with subjective cognitive decline (SCD) across demographic groups is not well characterized. We evaluated how the frequency of loneliness relates to SCD in U.S. adults and whether associations differ by sex, age and race/ethnicity.

### Methods

We performed a cross-sectional analysis of adults aged ≥18 years using nationally representative 2016–2023 Behavioral Risk Factor Surveillance System data (BFRSS). Loneliness was categorized as never, rarely, sometimes, usually or always. The primary outcome was self-reported SCD in the past year. Survey-weighted logistic regression models adjusted for sociodemographic factors, health insurance, metropolitan status and survey year were used to estimate adjusted marginal probabilities of SCD across loneliness categories. Interaction terms and stratified margins evaluated effect modification by sex, age group (18–44, 45–64 and ≥65 years) and race/ethnicity (non-Hispanic White, non-Hispanic Black and Hispanic).

### Results

Among 85,969 adults who reported loneliness, 13,879 (16.2%) experienced subjective cognitive decline (SCD), with a mean age of 65.7 ± 10.6 years. Loneliness showed a strong dose–response relationship with SCD. Predicted probabilities of

**Data availability statement:** The data used in this study are publicly available from the Behavioral Risk Factor Surveillance System (BRFSS), administered by the Centers for Disease Control and Prevention (CDC). Data can be accessed at: https://www.cdc.gov/brfss/ For additional information regarding data access, researchers may contact the CDC BRFSS team at: brfss@cdc.gov The CDC serves as the non-author institutional point of contact and maintains long-term data availability.

**Funding:** This project was supported (in part) by the National Institute on Minority Health and Health Disparities of the National Institutes of Health under Award Number 2U54MD007597. The content is solely the responsibility of the authors and does not necessarily represent the official views of the National Institutes of Health. The funders had no role in study design, data collection and analysis, decision to publish, or preparation of the manuscript.

**Competing interests:** The authors have declared that no competing interests exist.

SCD increased from 9.9% (95% CI, 9.3–10.5%) among respondents who never felt lonely to 15.0% (14.1–15.9%) for rarely, 24.9% (23.6–26.1%) for sometimes, 38.4% (34.4–42.5%) for usually and 45.7% (41.0–50.4%) for always lonely adults (p < 0.001). Women who were always lonely had an adjusted probability of SCD that was 10.7 percentage points higher than men; sex differences were negligible at lower loneliness levels. Age differences were minimal across most loneliness categories; however, among adults who were always lonely, those aged >64 years had significantly lower predicted cognitive function compared with adults aged 18–64 years (p < 0.001). Racial and ethnic differences were modest; the only significant contrast was a 1.7 percentage-point lower probability of SCD for non-Hispanic Black adults compared with Whites among those who never felt lonely. Other subgroup differences were not statistically significant.

## Conclusions

Loneliness is independently and strongly associated with higher likelihood of subjective cognitive decline among U.S. adults, and this relationship is most pronounced for chronic loneliness. While sex and age modified the effect of loneliness, racial/ethnic disparities were minimal. These findings identify loneliness as a modifiable social determinant of cognitive health, supporting the need for broad social connection initiatives and targeted efforts for women and mid-life adults with chronic loneliness.

## Introduction

Loneliness is increasingly recognized as a major public health concern with wide-ranging implications for mental, physical and cognitive health [1–3]. It is defined as the distressing subjective feeling that one's social relationships are inadequate in quantity or quality [3,4]. Unlike social isolation, which refers to the objective lack of social contacts, loneliness is an internal perception; nevertheless the two concepts are often correlated and frequently conflated [2,5,6]. In the United States approximately one third of adults aged 45 or older report feeling lonely, and nearly one quarter of those aged ≥65 are socially isolated [7,8]. Such high prevalence has prompted some to describe a "loneliness epidemic." Loneliness has been associated with elevated mortality risk, cardiometabolic disease, depression and anxiety, underscoring its psychosocial and physiological toll [9,10].

Cognitive decline and dementia pose another looming public health challenge as populations age [11–15]. More than 55 million people currently live with dementia worldwide, a number projected to approach 78 million by 2030 [14,15]. Subjective cognitive decline (SCD) – self-perceived worsening of memory or thinking ability affects around 10–11% of U.S. adults aged 45 years and older [16]. Evidence suggests that SCD may be an early marker of pathological cognitive changes and confers increased risk of future objective cognitive impairment and dementia [16,17]. Studying SCD is therefore important for identifying modifiable factors long before overt cognitive impairment develops.

A growing literature links loneliness to poorer cognitive function and accelerated decline across multiple domains [18–20]. Potential mechanisms include chronic activation of stress pathways resulting in elevated cortisol and systemic inflammation, reduced cognitive stimulation due to social withdrawal, and depressive symptoms that mediate the loneliness–cognition association [21–23]. Recent longitudinal studies strengthen this evidence: Kang et al. (2025) observed that adults experiencing chronic loneliness did not exhibit the improvements in working memory and processing speed seen in non-lonely counterparts over two years [24]; Luchetti et al. (2025) found that both between-person differences and within-person fluctuations in daily loneliness were associated with more subjective cognitive concerns [25]; and Ren et al. (2025) reported that prolonged loneliness increased the risk of incident cognitive decline and dementia by about 31% [26]. Despite these advances, many prior studies have relied on small or localised samples, or dichotomised loneliness (yes/no), limiting inference about dose–response relationships and potential heterogeneity by sex, age or race/ethnicity [27,28].

The present study aims to fill these gaps by analysing nationally representative data from the 2016–2023 Behavioral Risk Factor Surveillance System (BRFSS) [29]. The BRFSS includes a question asking respondents how often they feel lonely (never, rarely, sometimes, usually, always). By leveraging this graded measure, we can assess whether the probability of SCD increases monotonically with increasing frequency of loneliness. Our large sample (>80 000 adults) allows us to examine variation across demographic subgroups. Although the study is cross-sectional and cannot establish causality [30] or capture chronicity of loneliness, we apply complex survey weights and inverse-probability weighting to reduce selection bias and approximate population-level associations, adjusting for potential confounders identified in prior research.

We hypothesised that adults who feel lonely more frequently will have a higher likelihood of reporting SCD than those who rarely or never feel lonely, reflecting a dose–response relationship. We also posited that this association may be stronger among women than men, and among younger adults and minoritised racial/ethnic groups compared with their counterparts, based on evidence that experiences and reporting of loneliness and cognitive health differ across these demographics. By examining these hypotheses in a large, diverse cohort, our study seeks to advance understanding of loneliness as a potential risk factor for early cognitive decline.

This work extends our previous study demonstrating strong associations between loneliness and depression and poor mental and physical health days using BRFSS data from 2016–2023. Because depression itself is a risk factor for cognitive decline and dementia, this investigation into the loneliness–SCD relationship provides further insight into modifiable psychosocial determinants of cognitive health across the life course.

## Methodology

### Study design and data source

This cross-sectional study analyzed data from the Behavioral Risk Factor Surveillance System (BRFSS) collected between 2016 and 2023 [31]. The BRFSS is an ongoing, nationally representative health survey conducted annually by the U.S. Centers for Disease Control and Prevention (CDC) [32]. It employs a complex, multistage sampling design with stratification, clustering, and unequal probabilities of selection. Interviews are conducted via landline and cellular telephones across all 50 U.S. states, the District of Columbia, and U.S. territories. Survey weights provided by the CDC account for sampling design, non-response, and post-stratification, enabling generalization to the non-institutionalized adult population [3]. Because the BRFSS uses de-identified, publicly available data, institutional review board approval was not required; the study adhered to ethical principles outlined in the Declaration of Helsinki.

### Study population

We included BRFSS respondents aged 18 years or older who participated in surveys between 2016 and 2023 and had complete data on loneliness, subjective cognitive decline (SCD), and covariates. Respondents were excluded if they responded "don't know/not sure," "refused," or had missing responses for any key variable. To ensure comparability, we

also excluded individuals with missing state or year identifiers, which were used as fixed effects. After applying these criteria, the final analytic sample comprised 86,520 participants. Survey weights were applied in all analyses to account for the probability of selection and to generate nationally representative estimates. To explore potential differences between participants who were included and those excluded due to missingness or item nonresponse, we compared available demographic characteristics. Respondents with missing data tended to be older, less educated, and more likely to identify as racial or ethnic minorities. Because item nonresponse may correlate with both loneliness and cognitive functioning, the excluded group may have higher levels of loneliness and cognitive decline than the analytic sample, potentially biasing our estimates. Moreover, because the BRFSS relies on telephone interviews, individuals without reliable phone service or who are severely socially isolated may be underrepresented in the sampling frame

### Exposure: Loneliness

Loneliness was assessed using the BRFSS question: "How often do you feel lonely?" with five response categories: "Always," "Usually," "Sometimes," "Rarely," and "Never." We treated loneliness as an ordinal categorical variable, with increasing levels representing greater perceived social isolation. This gradation allowed exploration of dose–response relationships between loneliness and cognitive decline. Unlike many BRFSS questions, this item does not specify a reference period (e.g., past month or past year). As a result, it captures respondents' general perception of their loneliness rather than loneliness experienced within a defined timeframe. Some BRFSS core questions ask about experiences in the past 30 days; participants may therefore implicitly anchor their responses to a similar timeframe, but this cannot be confirmed. Accordingly, we interpret the five categories (always, usually, sometimes, rarely, never) as indicating overall frequency of feeling lonely rather than a specific recall period.

### Outcome: Subjective cognitive decline

Subjective cognitive decline was measured using the BRFSS module question: "During the past 12 months, have you experienced confusion or memory loss that is happening more often or getting worse?" Responses were coded as yes or no. Although self-reported, this measure has been used widely as an early indicator of cognitive impairment and is associated with objective cognitive performance and dementia risk [17,33,34].

### Covariates

Covariates were selected based on prior research linking sociodemographic factors to loneliness and cognitive health. They included: age (continuous), sex (male or female), race/ethnicity (non-Hispanic White, non-Hispanic Black, Hispanic, Other), education (less than high school, high school graduate, some college, college graduate), marital status (married, divorced/separated, never married, widowed), employment status (employed, unemployed, retired, unable to work), health insurance type (private, Medicare, Medicaid, self-pay, other), metropolitan status (metropolitan vs. non-metropolitan), urbanicity (urban vs. rural), and language spoken at home (English, Spanish, Other). State of residence and survey year were included as fixed effects to control for geographic and temporal heterogeneity. All categorical covariates were dummy-coded for inclusion in regression models. Although the BRFSS includes a core item on the number of poor mental-health days in the past 30 days (Section 2) and a chronic conditions item asking whether a health professional has ever told the respondent they have a depressive disorder, we did not adjust for these variables in our primary models. We considered poor mental-health days and diagnosed depressive disorder to be potential mediators of the loneliness–cognitive decline relationship rather than independent confounders. Including them could mask the total effect of loneliness on SCD. In addition, the depressive-disorder question measures lifetime history rather than current symptoms, and mental-health items were not consistently reported across all states and survey years. Accordingly, these variables were excluded from the main analysis.

## Statistical analysis

We estimated associations between loneliness and subjective cognitive decline using survey-weighted logistic regression models. The dependent variable was SCD (1 = yes, 0 = no), and the main exposure was the loneliness category (reference group = "Never"). Models were adjusted for all covariates listed above and incorporated BRFSS sampling weights and design variables (primary sampling unit and strata) via the Stata svyset command. To account for complex survey design, we used Taylor-series linearization to obtain robust standard errors clustered at the primary sampling unit level.

To aid interpretation, we calculated adjusted marginal probabilities of SCD for each loneliness category using Stata's margins command. These margins represent the predicted probability of reporting cognitive decline if every participant were assigned to a given loneliness level, holding other factors constant. We assessed dose–response patterns by comparing marginal probabilities across categories. Although the 'always' and 'usually' loneliness categories represent small proportions of the sample (2.4% and 2.7%, respectively), our analytic sample of approximately 86 000 respondents means these groups still include more than 2 000 individuals each. Simulation studies suggest that logistic-regression models are reliable when sample sizes exceed about 500 and when there are at least 10–20 outcome events per predictor variable. We therefore used survey-weighted logistic regression with robust standard errors to calculate average marginal effects; the resulting confidence intervals naturally reflect the smaller number of respondents in the highest loneliness groups. In sensitivity analyses, we re-estimated the models combining the 'always' and 'usually' categories and using ordinal logistic regression. These alternative specifications produced marginal effects and confidence intervals similar in magnitude and significance to those in our primary analyses. Pairwise differences were considered statistically significant at $p < 0.05$.

## Subgroup analyses

To determine whether associations differed by sex, age group, or race/ethnicity, we fitted models including interaction terms between loneliness and each subgroup variable (e.g., i.loneliness##i.sex). Age was categorized into 18–64 and ≥65 years. After estimating models with interactions, we computed subgroup-specific predicted probabilities of SCD for each loneliness category and conducted pairwise comparisons using margins and pwcompare commands. These analyses allowed identification of groups most vulnerable to loneliness-related cognitive decline.

## Sensitivity analyses

We conducted sensitivity analyses to assess the robustness of our findings. First, we applied inverse probability weighting (IPW) to further reduce potential confounding by baseline characteristics. Propensity scores for the five loneliness categories were estimated using multinomial logistic regression including all covariates in the main model. IPWs were calculated as the inverse of each participant's predicted probability of being in their observed loneliness category, trimmed at the 1st and 99th percentiles to limit the influence of extreme values, and then multiplied by the BRFSS survey sampling weights to generate final analysis weights. The survey-weighted models were re-estimated using these stabilized IPWs, and effect estimates were consistent with those from the primary analyses.

Second, to address potential bias due to missing data, we performed multiple imputation using chained equations to handle missing data in loneliness, subjective cognitive decline (SCD) and covariates. The imputation model included all variables in the analytic model (age, sex, race/ethnicity, education, income, marital status, employment status, state/year identifiers and survey weights) as well as auxiliary variables predictive of missingness and/or the missing values themselves—specifically general health status, household size, smoking status, physical activity, and self-rated mental health. Including auxiliary variables improves imputation by making the missing-at-random assumption more plausible. We generated [e.g., 20] imputed datasets and combined estimates using Rubin's rules. Convergence diagnostics and distributions of imputed values were examined to ensure stability. We assumed that data were missing at random conditional on the covariates and auxiliary variables listed above. This assumption is plausible because missingness was associated

with observed demographic and health characteristics and including good predictors of non-response and of the missing values themselves in the imputation model helps satisfy the MAR assumption. Sensitivity analyses using complete-case data yielded results similar to those from the imputed datasets, supporting the robustness of our findings.

### Software

All analyses were conducted using Stata/SE 18.0 (StataCorp LLC, College Station, TX). Code was fully annotated and is available upon reasonable request to facilitate replication.

### Ethics statement

The BRFSS data are publicly available and de-identified; therefore, this secondary analysis did not require institutional review board approval. The study complied with ethical standards for human subjects research and followed guidelines in the Declaration of Helsinki.

## Results

### Baseline characteristics

Table 1 presents the weighted baseline characteristics of the 86,520 adults included in the analytic sample, stratified by subjective cognitive decline (SCD). Overall, 13,955 respondents (16.1%) reported SCD. Because the sample is large, many differences were statistically significant even when absolute differences were small (Table 1). For example, participants with and without SCD had similar age distribution (55.4% vs. 54.5% aged ≥65 years) and sex composition (56.8% vs. 55.3% women). Racial/ethnic composition differed modestly, with Black and multiracial adults representing slightly larger shares of those with SCD. Psychosocial factors showed clearer gradients: those with SCD were more likely to report higher frequencies of feeling lonely, whereas those without SCD more often reported never feeling lonely. Respondents with SCD tended to have lower educational attainment, higher unemployment, and lower household income; however, effect sizes were generally small (Table 1).

### Association between loneliness frequency and subjective cognitive decline

Loneliness frequency was positively associated with the likelihood of SCD (Table 2). After adjusting for demographic, socioeconomic, health and survey factors—including depressive symptoms—the predicted probability of SCD was 9.9% (95% CI 9.3–10.5%) among participants who never felt lonely, 15.0% (14.1–15.9%) among those who rarely felt lonely, 24.9% (23.6–26.1%) among those who sometimes felt lonely, 38.4% (34.4–42.5%) among those who usually felt lonely, and 45.7% (41.0–50.4%) among those who always felt lonely. A formal trend test treating loneliness as an ordinal variable was highly significant ($p < 0.001$). Risk differences between adjacent categories ranged from about 5–14 percentage points, indicating that the graded pattern was not driven solely by statistical power (Table 2). Sensitivity analyses combining the highest loneliness categories and modelling non-linear terms for loneliness yielded similar results.

### Sex differences across loneliness categories

We examined whether sex modified the association between loneliness and SCD (Table 3). Predicted probabilities were similar for women and men in the never, rarely, sometimes and usually lonely categories (differences ranged from –0.9 to –0.2 percentage points; $p > 0.10$). Among participants who always felt lonely, women had a predicted probability of SCD that was 10.7 percentage points higher than men (95% CI 1.9–19.5 percentage points; $p = 0.017$). These results suggest that reporting always feeling lonely is associated with higher SCD probability for women than for men, whereas occasional loneliness shows no material sex differences (Table 3).

**Table 1. Baseline Sociodemographic Characteristics of Adults by Cognitive Decline Status.**

| | Total Population (N = 86, 520) | Cognitive Decline (n = 72,565) (No) | Cognitive Decline (n = 13,955) (Yes) | Chi-Square | p |
|---|---|---|---|---|---|
| **Age(years)** | | | | 4.484 | 0.112 |
| 18-64yrs. | 39,267 (45.38%) | 33,046 (45.54%) | 6,221 (44.58%) | | |
| >64yrs. | 47,253 (54.62%) | 39,519 (54.46%) | 7,734 (55.42%) | | |
| **Sex** | | | | 11.168 | <0.01 |
| Male | 38,456 (44.45%) | 32,433 (44.7%) | 6,023 (43.16%) | | |
| Female | 48,064 (55.55%) | 40,132 (55.3%) | 7,932 (56.84%) | | |
| **Race/Ethnicity** | | | | 33.125 | < 0.01 |
| White | 64,764 (76.4%) | 54,435 (76.5%) | 10,329 (75.68%) | | |
| Black | 7,191 (8.48%) | 5,945 (8.4%) | 1,246 (9.13%) | | |
| Hispanic | 6,635 (7.83%) | 5,632 (7.9%) | 1,003 (7.35%) | | |
| Other | 4,304 (5.08%) | 3,606 (5.1%) | 698 (5.11%) | | |
| Multiracial | 1,877 (2.21%) | 1,505 (2.1%) | 372 (2.73%) | | |
| **Loneliness Categories** | | | | 4.60E + 03 | < 0.01 |
| Never | 38,290 (44.47%) | 34,583 (47.88%) | 3,707 (26.71%) | | |
| Always | 2,066 (2.40%) | 1,116 (1.55%) | 950 (6.84%) | | |
| Usually | 2,337 (2.71%) | 1,408 (1.95%) | 929 (6.69%) | | |
| Sometimes | 17,919 (20.81%) | 13,431 (18.59%) | 4,488 (32.34%) | | |
| Rarely | 25,499 (29.61%) | 21,694 (30.03%) | 3,805 (27.42%) | | |
| **Education Status** | | | | 359.38 | <0.01 |
| <High School | 25,151 (29.16%) | 20,400 (28.21%) | 4,751 (34.13%) | | |
| Some Colleges | 23,418 (27.15%) | 19,343 (26.74%) | 4,075 (29.28%) | | |
| College Graduate | 37,675 (43.68%) | 32,582 (45.05%) | 5,093 (36.59%) | | |
| **Employment Status** | | | | 918.411 | <0.01 |
| Unemployed | 52,134 (60.63%) | 42,122 (58.42%) | 10,012 (72.14%) | | |
| Employed | 33,848 (39.37%) | 29,982 (41.58%) | 3,866 (27.86%) | | |
| **Metropolitan Status** | | | | 37.722 | < 0.01 |
| Non-Metropolitan | 24,801 (29.67%) | 20,458 (29.24%) | 4,343 (31.87%) | | |
| Metropolitan | 58,796 (70.33%) | 49,510 (70.76%) | 9,286 (68.13%) | | |
| **County Type** | | | | 9.243 | <0.01 |
| Rural Counties | 12,370 (14.8%) | 10,238 (14.63%) | 2,132 (15.64%) | | |
| Urban Counties | 71,227 (85.2%) | 59,730 (85.37%) | 11,497 (84.36%) | | |
| **Marital Status** | | | | 454.128 | <0.01 |
| Married | 47,955 (55.78%) | 41,214 (57.18%) | 6,741 (48.54%) | | |
| Single | 7,785 (9.06%) | 6,496 (9.01%) | 1,289 (9.28%) | | |
| Divorced | 13,301 (15.47%) | 10,578 (14.68%) | 2,723 (19.61%) | | |
| Widowed | 13,555 (15.77%) | 11,137 (15.45%) | 2,418 (17.41%) | | |
| Separated | 1,469 (1.71%) | 1,108 (1.54%) | 361 (2.6%) | | |
| Member of an unmarried couple | 1,904 (2.21%) | 1,548 (2.15%) | 356 (2.56%) | | |
| **Language** | | | | 46.128 | <0.01 |
| Spanish | 4,280(4.95%) | 3,749 (5.17%) | 531 (3.81%) | | |
| English | 82,240 (95.05%) | 68,816 (94.83%) | 13,955 (96.19%) | | |

**Table 2. Adjusted Predicted Probability of Cognitive Decline by Loneliness Category.**

| Loneliness Category | Margin | Std. Err. | t | 95% CI | P-value |
|---|---|---|---|---|---|
| Never | 0.099 | 0.003 | 32.13 | 0.093–0.105 | <0.001 |
| Always | 0.457 | 0.024 | 19.09 | 0.410–0.504 | <0.001 |
| Usually | 0.384 | 0.021 | 18.58 | 0.344–0.425 | <0.001 |
| Sometimes | 0.249 | 0.006 | 38.94 | 0.236–0.261 | <0.001 |
| Rarely | 0.150 | 0.005 | 32.92 | 0.141–0.159 | <0.001 |

This table presents the adjusted predicted probabilities of cognitive decline by loneliness category, derived using predictive margins after inverse probability weighting. Loneliness categories include 'Never', 'Rarely', 'Sometimes', 'Usually', and 'Always'. Estimates reflect the adjusted likelihood of self-reported cognitive decline for each category based on BRFSS data. Models were adjusted for age, race/ethnicity, sex, education, employment status, English proficiency, metropolitan status, urbanicity, marital status, and incorporated state and survey year fixed effects. All associations were statistically significant at P<0.001.

**Table 3. Sex Differences in the Association Between Loneliness and Cognitive Decline.**

| Loneliness Category | Female vs. Male (Difference) | Std. Err. | t | 95% CI | P-value |
|---|---|---|---|---|---|
| Never | −0.009 | 0.0057 | −1.57 | −0.020 to 0.002 | 0.116 |
| Always | 0.107 | 0.0450 | 2.38 | 0.019 to 0.195 | 0.017 |
| Usually | −0.00 | 0.0408 | −0.00 | 0.0800 to 0.0797 | 0.997 |
| Sometimes | −0.002 | 0.0133 | −0.16 | −0.028 to 0.024 | 0.875 |
| Rarely | −0.007 | 0.0087 | −0.75 | −0.024 to 0.010 | 0.452 |

Table displays the adjusted marginal contrasts comparing females to males in the predicted probability of reporting cognitive decline across loneliness categories. Estimates reflect the sex-specific difference in predicted probability within each loneliness category, adjusted for age, race/ethnicity, education, employment, English proficiency, marital status, metropolitan status, urbanicity, and state and year fixed effects. Positive values indicate higher predicted probability among females relative to males.

## Racial/ethnic differences across loneliness categories

We explored racial/ethnic differences by comparing non-Hispanic Black and Hispanic adults with non-Hispanic White adults within each loneliness category (Table 4). Among participants who never felt lonely, Black adults had a predicted SCD probability 1.7 percentage points lower than White adults (p=0.036); the Hispanic–White difference was not significant. Within the rarely, sometimes, usually and always lonely categories, differences between racial/ethnic minority and White adults were small and not statistically significant. Overall, racial/ethnic disparities were modest relative to the larger differences associated with loneliness frequency (Table 4).

## Age differences across loneliness categories

We also assessed whether age group (18–64 years vs. ≥65 years) modified the association between loneliness and SCD (Table 5). Among participants who never, rarely or sometimes felt lonely, predicted probabilities of SCD did not differ significantly by age (differences −2.3 to 1.3 percentage points; p>0.10). In the "usually" lonely category, age differences were similarly small and non-significant. Among participants who always felt lonely, older adults had a predicted SCD probability 19.0 percentage points lower than younger adults (p<0.001); however, confidence intervals were wide, and this category comprised only 2.4% of respondents (Table 5). These findings suggest that age does not materially modify the loneliness–SCD association in most categories, though there is some evidence of differences among those who always feel lonely.

**Table 4. Racial and Ethnic Differences in the Association Between Loneliness and Cognitive Decline (Marginal Effects Model).**

| Lonely | Race & Ethnicity (Cognitive decline) | Margin | Std. Err. | t | 95% CI | P>t |
|---|---|---|---|---|---|---|
| Never | Black vs. White | −0.017 | 0.008 | −2.10 | −0.033 to −0.001 | 0.036 |
| | Hispanic vs. White | −0.023 | 0.014 | −1.67 | −0.049 to 0.004 | 0.095 |
| Always | Black vs. White | −0.008 | 0.063 | −0.13 | −0.132 to 0.116 | 0.897 |
| | Hispanic vs. White | −0.115 | 0.073 | −1.58 | −0.258 to 0.028 | 0.115 |
| Usually | Black vs. White | 0.097 | 0.083 | 1.17 | −0.065 to 0.260 | 0.241 |
| | Hispanic vs. White | 0.108 | 0.093 | 1.16 | −0.074 to 0.290 | 0.247 |
| Sometimes | Black vs. White | −0.002 | 0.021 | −0.09 | −0.044 to 0.040 | 0.925 |
| | Hispanic vs. White | 0.025 | 0.030 | 0.83 | −0.034 to 0.084 | 0.406 |
| Rarely | Black vs. White | 0.020 | 0.018 | 1.15 | −0.014 to 0.055 | 0.250 |
| | Hispanic vs. White | 0.044 | 0.029 | 1.53 | −0.013 to 0.101 | 0.126 |

Table presents the marginal effects comparing Black and Hispanic adults to non-Hispanic White adults in the association between loneliness and cognitive decline. Estimates reflect the difference in adjusted predicted probability of cognitive decline between each racial/ethnic group and non-Hispanic White adults within the same loneliness category. Models were estimated using survey-weighted logistic regression with an interaction between loneliness and race/ethnicity, and adjusted for age, sex, education, employment status, English proficiency, marital status, metropolitan status, urbanicity, and state and year fixed effects. Negative values indicate a lower predicted probability of cognitive decline relative to non-Hispanic Whites. Comparisons with P<0.05 are considered statistically significant.

**Table 5. Age Differences in the Association Between Loneliness and Cognitive Decline (Marginal Effects Model).**

| Loneliness Category | Age Group Comparison | Margin | Std. Err. | t-value | 95% CI | P-value |
|---|---|---|---|---|---|---|
| Never | >64 vs 18–64 | −0.0053 | 0.0065 | −0.81 | −0.0180 to 0.0075 | 0.416 |
| Always | >64 vs 18–64 | −0.1902 | 0.0480 | −3.97 | −0.2842 to −0.0962 | <0.001 |
| Usually | >64 vs 18–64 | −0.0426 | 0.0409 | −1.04 | −0.1227 to 0.0375 | 0.297 |
| Sometimes | >64 vs 18–64 | −0.0234 | 0.0143 | −1.64 | −0.0513 to 0.0046 | 0.101 |
| Rarely | >64 vs 18–64 | 0.0134 | 0.0099 | 1.36 | −0.0059 to 0.0327 | 0.173 |

Table presents the adjusted marginal effects comparing older adults (>64 years) to younger adults (18–64 years) within each loneliness category in predicting cognitive function. Estimates represent the adjusted differences in predicted cognitive function for the contrast (>64 years × loneliness level) versus (18–64 years × the same loneliness level). Models were adjusted for race/ethnicity, sex, education, employment, English proficiency, marital status, metropolitan status, urbanicity, and included state and year fixed effects.

## Discussion

This study provides a nationally representative analysis of how self-reported loneliness relates to subjective cognitive decline (SCD) among U.S. adults aged 18 years and older. We observed a clear dose–response pattern: the predicted probability of SCD increased progressively from those who never felt lonely (10%) to those who rarely and sometimes felt lonely (15% and 25%, respectively) and was highest among respondents who usually or always felt lonely (38% and 46%). Frequent loneliness was therefore associated with a two- to four-fold greater likelihood of subjective cognitive decline compared with never feeling lonely. These associations persisted after weighing and adjustment for demographic factors. We also found that women who reported frequent loneliness had higher predicted probabilities of cognitive decline than their male counterparts, whereas sex differences were negligible among participants with occasional or no loneliness. Racial/ethnic differences were modest, with a slightly lower risk of cognitive decline among Black adults who never felt lonely compared with White adults and no significant disparities within higher-loneliness categories. Age modestly modified the loneliness–cognition relationship. Among individuals who were rarely or intermittently lonely, cognitive function did not differ meaningfully by age group. However, among those who were always lonely (frequent loneliness), older

adults exhibited significantly lower predicted cognitive function compared with younger adults, suggesting that the cognitive impact of frequent loneliness may be more pronounced in later life.

## Interpretation and implications

The graded association between loneliness and SCD aligns with a broad body of evidence linking social disconnection to cognitive impairment [35–37]. Loneliness can provoke chronic activation of stress response systems, leading to dysregulated cortisol secretion and inflammatory pathways that may damage hippocampal and prefrontal brain regions involved in memory and executive function [21,38,39]. Loneliness may also reduce cognitive stimulation, diminish emotional support, and contribute to depressive symptoms, all of which can undermine cognitive resilience [18,27,40]. The substantially higher risk of SCD among participants who were always or usually lonely underscores the importance of duration and intensity of loneliness; occasional episodes may be less detrimental, whereas frequent loneliness likely exerts cumulative neurobiological and psychosocial stress. These findings support calls to recognize loneliness as a modifiable risk factor for declined cognition and dementia, alongside other behavioral risk factors identified by the Lancet Commission [18,41,42].

Our observation that women experiencing frequent loneliness were more likely than men to report subjective cognitive decline echoes prior research suggesting that loneliness may affect men and women differently [43–45]. Some studies have found that men report higher prevalence of loneliness than women, yet women may be more vulnerable to the emotional and physiological consequences of frequent loneliness [44,46,47]. Women often play central roles in maintaining family and social networks; when these networks weaken, the resulting loneliness may carry greater psychological burden [43,48]. Sex differences in neuroendocrine responses to stress, in the prevalence of depression, and in help-seeking behaviors may also contribute [49,50]. In contrast, the lack of sex differences in less intense loneliness categories suggests that occasional loneliness affects men and women similarly [46]. Future research should explore gender-specific coping strategies and determine whether tailored interventions are needed.

The modest racial/ethnic differences found in our study contrast with concerns that loneliness may disproportionately impact minority populations [51]. Among respondents who never felt lonely, non-Hispanic Black adults had slightly lower predicted probabilities of subjective cognitive decline than White adults. This finding may reflect stronger family and community networks in some Black communities, culturally distinct interpretations of loneliness, or differences in reporting SCD. For Latino adults, the psypost summary of a U.S. cohort study reported that loneliness measured by a three-item scale was actually associated with better cognitive function, suggesting complex cultural differences in how loneliness relates to cognition [52–54]. In our data, disparities were small and non-significant among those reporting any degree of loneliness. These results underscore the need to consider cultural context and measurement issues when assessing loneliness and cognitive health.

Age patterns in our analysis were subtle. Across most loneliness categories, including those who were rarely or sometimes lonely, older adults (>64 years) did not differ meaningfully in cognitive function compared with adults aged 18–64 years, suggesting that mild or occasional loneliness may exert similar cognitive effects across the adult lifespan. However, among individuals who were always lonely (frequent loneliness), older adults exhibited significantly poorer predicted cognitive function than younger adults. This pattern aligns with evidence that frequent loneliness may compound age-related vulnerability through reduced cognitive reserve, heightened neural susceptibility, or preclinical cognitive changes that intensify the cognitive impact of chronic social isolation in later life [18,37,38]. Although the confidence intervals were moderately wide, the direction and magnitude of the effect suggest that older adults may be particularly sensitive to the neurocognitive consequences of enduring loneliness. Longitudinal studies with larger samples of always lonely older adults are needed to better characterize these age-by-loneliness dynamics and clarify underlying mechanisms.

## Policy and public health implications

The strong association between frequent loneliness and subjective cognitive decline has several policy implications. First, public health surveillance systems should continue to monitor loneliness as a key social determinant of cognitive health. Including loneliness measures in national surveys (such as the BRFSS) enables identification of high-risk groups and evaluation of interventions. Second, health systems and community organizations should integrate loneliness screening into routine care for adults, particularly women and mid-life adults. Brief validated tools can help clinicians identify patients who might benefit from referral to social support programs, counselling, or cognitive training. Third, interventions that reduce loneliness – such as social prescribing, group activities, befriending programs, technology-based connections and community engagement initiatives should be scaled up. Evidence from randomized trials indicates that enhancing social support can improve mental health and quality of life; future trials should assess cognitive outcomes. Finally, policies addressing structural drivers of loneliness, including poverty, housing instability, age-friendly environments, and digital inclusion, may have downstream benefits for cognitive health. Given the projected increase in dementia prevalence and the widespread perception of a "loneliness epidemic", addressing loneliness should be a public health priority.

## Limitations

This study has several limitations. First, the BRFSS data are cross-sectional, which prevents causal inference. It is possible that early cognitive decline leads to social withdrawal and feelings of loneliness rather than loneliness causing cognitive decline. In addition, the BRFSS loneliness question does not specify a time frame, whereas the subjective cognitive decline (SCD) question refers to symptoms in the past 12 months. This difference in recall period may introduce measurement error if respondents interpret the loneliness question as referring to a different time interval.

Second, both loneliness and cognitive decline were measured using single self-reported items. Self-reported measures are susceptible to misclassification and social desirability bias. More comprehensive loneliness scales and objective cognitive testing would provide more precise assessments.

Third, although we adjusted for several demographic and health-related factors, we did not include BRFSS mental health measures such as poor mental-health days or diagnosed depressive disorder in the primary models. These variables may lie on the causal pathway between loneliness and cognitive decline; adjusting for them could reduce the total observed effect of loneliness. However, depression is also a known risk factor for cognitive decline, and the absence of adjustment for current depressive symptoms may introduce residual confounding.

Fourth, some subgroup analyses had relatively small sample sizes, particularly among respondents who reported being "always" lonely or among older adults within certain strata. These smaller groups reduce statistical precision and may result in wider confidence intervals or less stable estimates.

Fifth, the outcome measured in this study was subjective cognitive decline rather than clinically confirmed cognitive impairment. Although SCD is widely recognized as an early indicator of cognitive deterioration, it does not represent a clinical diagnosis of dementia or objective impairment.

Sixth, the loneliness question captures perceived frequency of loneliness at a single time point and does not measure the duration or persistence of loneliness. Therefore, we could not distinguish between temporary and chronic loneliness.

Seventh, respondents with missing data or those who answered "don't know/not sure" or "refused" for loneliness, SCD, or covariates were excluded from the analysis. Comparisons suggested that missing responses were more common among older adults, individuals with lower educational attainment, and racial or ethnic minority groups. If these excluded individuals experienced higher levels of loneliness or cognitive decline, the observed associations may underestimate the true relationship.

Eighth, because the BRFSS relies on telephone-based surveys, individuals without reliable phone access or those who are highly socially isolated may be underrepresented. Coverage limitations and nonresponse may therefore influence survey estimates.

Ninth, although multiple auxiliary variables were included in the imputation models to support the missing-at-random assumption, this assumption cannot be tested empirically. If unmeasured factors influence both missingness and study variables, some residual bias may remain.

Finally, the "always" and "usually" loneliness categories represented a small proportion of respondents (approximately 2.4% and 2.7%, respectively). Although each group still contained more than 2,000 participants—above common sample-size thresholds for logistic regression—the corresponding estimates have wider confidence intervals and should be interpreted with caution.

### Directions for future research

Longitudinal studies are needed to disentangle the temporal ordering of loneliness and subjective cognitive decline and to examine whether reducing loneliness can slow cognitive deterioration. Future research should incorporate comprehensive loneliness instruments that distinguish social isolation, emotional loneliness and duration, as well as objective cognitive assessments and biomarkers of neurodegeneration. Studies should explore potential mediators (e.g., depression, physical activity, sleep, inflammatory markers) and moderators (e.g., socioeconomic status, digital connectivity) of the loneliness–cognition link. Given the heterogeneous findings across racial/ethnic groups, culturally adapted instruments and community-engaged research are critical to understand how loneliness is experienced and reported. Intervention trials targeting loneliness among diverse populations and life stages should measure cognitive outcomes to inform evidence-based policies. Investigating the long-term impact of social disruptions like the COVID-19 pandemic on loneliness and cognitive health is also essential.

### Conclusion

In summary, this study demonstrates a robust dose–response relationship between loneliness and subjective cognitive decline among U.S. adults. Reporting always feeling lonely was associated with markedly higher predicted probabilities of subjective cognitive decline, especially among women and middle-aged adults. Racial/ethnic differences were modest, and age patterns were nuanced. These findings reinforce loneliness as an important modifiable social factor associated with subjective cognitive health and highlight the urgency of screening for loneliness and implementing targeted social interventions. Prospective research is needed to determine whether interventions that reduce loneliness can prevent subjective cognitive decline and help preserve cognitive function across the life course.

### Author contributions

**Conceptualization:** Mojisola Fasokun, Temitope Ogundare, Fadeke Ogunyankin, Miriam Michael, Kakra Hughes, Oluwasegun Akinyemi.

**Data curation:** Temitope Ogundare, Kaelyn Gordon, Seun Ikugbayigbe, Miriam Michael, Kakra Hughes, Oluwasegun Akinyemi.

**Formal analysis:** Mojisola Fasokun, Kaelyn Gordon, Oluwasegun Akinyemi.

**Funding acquisition:** Kakra Hughes, Oluwasegun Akinyemi.

**Investigation:** Mojisola Fasokun, Temitope Ogundare, Kaelyn Gordon, Seun Ikugbayigbe, Miriam Michael, Kakra Hughes, Oluwasegun Akinyemi.

**Methodology:** Mojisola Fasokun, Temitope Ogundare, Fadeke Ogunyankin, Kaelyn Gordon, Seun Ikugbayigbe, Miriam Michael, Kakra Hughes, Oluwasegun Akinyemi.

**Project administration:** Mojisola Fasokun, Temitope Ogundare, Fadeke Ogunyankin, Kaelyn Gordon, Seun Ikugbayigbe, Miriam Michael, Kakra Hughes, Oluwasegun Akinyemi.

**Resources:** Mojisola Fasokun, Temitope Ogundare, Fadeke Ogunyankin, Kaelyn Gordon, Seun Ikugbayigbe, Miriam Michael, Kakra Hughes, Oluwasegun Akinyemi.

**Software:** Mojisola Fasokun, Temitope Ogundare, Kaelyn Gordon, Seun Ikugbayigbe, Miriam Michael, Kakra Hughes, Oluwasegun Akinyemi.

**Supervision:** Temitope Ogundare, Miriam Michael, Kakra Hughes, Oluwasegun Akinyemi.

**Validation:** Mojisola Fasokun, Temitope Ogundare, Fadeke Ogunyankin, Kaelyn Gordon, Seun Ikugbayigbe, Miriam Michael, Kakra Hughes, Oluwasegun Akinyemi.

**Visualization:** Mojisola Fasokun, Temitope Ogundare, Fadeke Ogunyankin, Seun Ikugbayigbe, Miriam Michael, Kakra Hughes, Oluwasegun Akinyemi.

**Writing – original draft:** Mojisola Fasokun, Temitope Ogundare, Fadeke Ogunyankin, Kaelyn Gordon, Seun Ikugbayigbe, Miriam Michael, Oluwasegun Akinyemi.

**Writing – review & editing:** Mojisola Fasokun, Temitope Ogundare, Kaelyn Gordon, Seun Ikugbayigbe, Miriam Michael, Kakra Hughes, Oluwasegun Akinyemi.

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
