## [Decision Letter · Decision Letter 0]

3 Feb 2026

PONE-D-25-65508Loneliness and Cognitive Decline Among U.S. Adults: A Stratified Analysis of the BRFSSPLOS One

Dear Dr. Fasokun,

Thank you for submitting your manuscript to PLOS ONE. After careful consideration, we feel that it has merit but does not fully meet PLOS ONE’s publication criteria as it currently stands. Therefore, we invite you to submit a revised version of the manuscript that addresses the points raised during the review process.

The manuscript has been evaluated by two reviewers, and their comments are available below.

The reviewers have raised a number of major concerns. They feel the manuscript requires substantial revisions to the text, as well as of the interpretation of the findings. They also request several clarifications regarding the methods.

Could you please carefully revise the manuscript to address all comments raised?

We look forward to receiving your revised manuscript.

Kind regards,

Alejandro Torrado Pacheco, PhD

Staff Editor

PLOS One

Journal Requirements:

“This project was supported (in part) by the National Institute on Minority Health and Health Disparities of the National Institutes of Health under Award Number 2U54MD007597. The content is solely the responsibility of the authors and does not necessarily represent the official views of the National Institutes of Health.”

3. In this instance it seems there may be acceptable restrictions in place that prevent the public sharing of your minimal data. However, in line with our goal of ensuring long-term data availability to all interested researchers, PLOS’ Data Policy states that authors cannot be the sole named individuals responsible for ensuring data access (http://journals.plos.org/plosone/s/data-availability#loc-acceptable-data-sharing-methods).

Before we proceed with your manuscript, please also provide non-author contact information (phone/email/hyperlink) for a data access committee, ethics committee, or other institutional body to which data requests may be sent. If no institutional body is available to respond to requests for your minimal data, please consider if there any institutional representatives who did not collaborate in the study, and are not listed as authors on the manuscript, who would be able to hold the data and respond to external requests for data access? If so, please provide their contact information (i.e., email address). Please also provide details on how you will ensure persistent or long-term data storage and availability

Reviewers' comments:

Reviewer's Responses to Questions

Comments to the Author

1. Is the manuscript technically sound, and do the data support the conclusions?

Reviewer #1: Partly

Reviewer #2: Yes

2. Has the statistical analysis been performed appropriately and rigorously? 

Reviewer #1: I Don't Know

Reviewer #2: No

3. Have the authors made all data underlying the findings in their manuscript fully available?

Reviewer #1: Yes

Reviewer #2: Yes

4. Is the manuscript presented in an intelligible fashion and written in standard English?

Reviewer #1: Yes

Reviewer #2: Yes

5. Review Comments to the Author

Reviewer #1: Thank you for the opportunity to read this interesting and important paper that investigated the relationship between the frequency of loneliness and subjective cognitive decline. The authors found that the frequency of loneliness was associated with subjective cognitive decline, showing predicted probability of SCD of 9.9% in respondents that never felt lonely, all the way up to 45% in those that always felt lonely. The paper is well written, clearly communicates study methods, and adds valuable insights into the broader literature on impacts and significance of social disconnection and demonstrates a thoughtful analysis to support whether interventions could be worthwhile. There are a few comments that the authors could address to strengthen the piece.

Major comments:

1.) The language used in certain parts of the paper needs adjustment.

- The study conclusions, "These findings identify loneliness as a modifiable social determinant of cognitive health" could be more cautiously and grounded in study findings

- The claim that a one-time cross-sectional assessment of loneliness describes chronicity of loneliness is a big claim, I believe the authors are limited since BRFSS doesn't include longitudinal data that would be more appropriate to look at trends for individuals rather than 1 time assessment of "chronicity". The primary difference between frequency and chronicity is in their focus: frequency measures how often an event occurs within a specific period, while chronicity measures how long a condition persists over a long-term, sustained period, there is a slight difference given what data is available and the wording should reflect this.

-There are several instances through the manuscript that outcome of "cognitive decline" is referred to without listing "subjective", it is important to include subjective throughout to minimize confusion, consider including in title and tables as well as text.

2.) Respondents that were excluded from the analyses because of "don't know, refused or missing key variable"- team may want to describe who is missing from their analysis. It is also possible that individuals with greatest loneliness frequency do not make it into a study like BRFSS and this limitation could be acknowledged.

3.) The authors clearly communicate study methods and it makes sense the use of adjusted marginal probabilities, but unclear if best way to assess dose response would be pairwise comparisons rather than trend test, and if pairwise is used then would consider the need for multiple pairwise comparisons correction to control the likelihood of false positives.

4.) The authors appropriately completed sensitivity analyses, but do not state which auxillary variables were included in their imputation models and this should be clarified as well as justification that missing at random assumption was met.

5.) One area that requires more consideration and explanation, Table 1 presents total population 86K and proportion across loneliness categories, only 2.4% of population always lonely and only 2.71% usually lonely (while Never lonely is ~45% and sometimes ~21%, and rarely ~30%). Given small proportion of individuals in these categories there is concern when using these group to calculate average marginal effects that there could be significant errors in estimation due to high variance and small sample size bias. While the marginal effects themselves might not be biased, the precision of these estimates (the standard errors) is often severely compromised, leading to unreliable inferences.

6.) The authors previous paper explores associations between loneliness and mental health (Akinyemi O, Abdulrazaq W, Fasokun M, Ogunyankin F, Ikugbayigbe S, Nwosu U, Michael M, Hughes K, Ogundare T. The impact of loneliness on depression, mental health, and physical well-being. PLoS One. 2025 Jul 9;20(7):e0319311. doi: 10.1371/journal.pone.0319311. PMID: 40632698; PMCID: PMC12240311.), which is likely relevant to subjective cognitive decline, but this is not examined in these analyses, unclear why. The paper states that they" lacked information on potentially confounding factors such as depressive symptoms", although the BRFSS has these measures, this needs to be clarified.

A few additional minor comments:

1.) In the methods section of abstract age range listed is "16-66" but minimum age of those in study was 18, this should be clarified or fixed.

2.) Limitation of cross sectional studies listed in introduction seems erroneous as the current study is also cross sectional, authors may want to be reword/ get more specific that other studies have looked at loneliness as binary yes/no without considering frequency/dose which is what it seems they are trying to convey

3.) The BRFSS study question on how often participants feel lonely does not include a timeframe, like over the past X amount of time, so it's unclear how participants may interpret this question, while question regarding SCD has clear time delineation of 12 months. Most other BRFSS study questions consider 30 day timeframe so its possible that is assumed by the participants, this could be conveyed more clearly for readers that are not familiar with BRFSS

4.) Methods section shares covariates were selected 'based on prior research', authors should include reference for what is being referred to.

5.) Some additional more recent references to consider that could be used to situate this current study in the discussion include:

- Kang JE, Martire LM, Graham-Engeland JE, Almeida DE, Sliwinski MJ. Chronic loneliness and longitudinal changes in cognitive functioning. BMC Public Health. 2025 Mar 29;25(1):1190. doi: 10.1186/s12889-025-22313-2. PMID: 40155901; PMCID: PMC11954266.

- Luchetti M, Aschwanden D, Stephan Y, Karakose S, Milad E, Miller AA, Zavala D, Hajek A, Terracciano A, Sutin AR. Loneliness and subjective cognitive concerns in daily life. Aging Ment Health. 2025 Oct;29(10):1856-1864. doi: 10.1080/13607863.2025.2519672. Epub 2025 Jun 20. PMID: 40539421; PMCID: PMC12321047.

- Ren Z, Luo Y, Liu Y, Gao J, Liu J, Zheng X. Prolonged loneliness and risk of incident cognitive decline and dementia: A two-cohort study. J Affect Disord. 2025 Jun 1;378:254-262. doi: 10.1016/j.jad.2025.03.001. Epub 2025 Mar 5. PMID: 40044082.

Reviewer #2: Thank you for the opportunity to review this manuscript.

Introduction

The Introduction would benefit from revisions to improve clarity, logical flow, and conceptual grounding across paragraphs. Here are some minor and major points:

1. Sentence structure and flow (Lines 57–58):

The sentence “Defined as a distressing subjective state arising when perceived social connections are inadequate” is a fragment and should be rewritten as a complete sentence or integrated with the preceding sentence.

2. Conceptual clarity between loneliness and social isolation (Lines 59–62):

The Introduction moves abruptly from prevalence estimates of social isolation (e.g., “nearly one quarter of those aged 65 and older are socially isolated”) to stating that loneliness and social isolation are distinct, followed immediately by “the present study focuses on loneliness.” This transition should be smoother.

3. Psychosocial vs. physiological framing (Lines 63–65):

The phrase “Beyond its psychosocial toll” implies that psychosocial consequences of loneliness have already been discussed, but they are not described earlier in the paragraph. This contrast should be clarified or rephrased to maintain logical continuity.

4. Insufficient contextualization of subjective cognitive decline (Lines 69–70):

Subjective cognitive decline is introduced briefly via prevalence estimates but is not conceptually developed. Given that cognitive decline, dementia, and SCD are related but distinct constructs, the authors should provide additional background on what SCD indicates, what it predicts, and why it is important to study SCD separately—particularly in relation to loneliness.

5. Blurring of background and methods (Lines 85–87):

The description of the BRFSS SCD item reads as methodological detail and may be more appropriate for the Methods section, or should be reframed as background information motivating the study.

6. Limited justification for analytic “contributions” (Lines 88–94):

The stated contributions would benefit from clearer theoretical or empirical justification, including: why a dose–response relationship between loneliness and SCD is hypothesized, how causal language is justified given the cross-sectional design, and why associations are expected to differ by sex, age, and race/ethnicity.

7. Need for more specific hypotheses (Lines 96–98):

The hypotheses are broad and largely restate the analytic plan. More specific, theory-driven hypotheses would strengthen the Introduction and better align with the proposed contributions.

Methods

1. Clearer justification for the inclusion of several covariates:

In particular, variables such as health insurance type, metropolitan status, urbanicity, and language appear to be included without explanation beyond their availability in the dataset. The authors should clarify the conceptual or empirical rationale for adjusting for these factors (e.g., whether they are considered confounders, proxies for access to care or socioeconomic context, or related to reporting of loneliness or subjective cognitive decline). Providing this justification would help readers assess the appropriateness of the adjustment set and improve interpretability of the findings.

Also, depressive symptoms/depression is a well-known factor of loneliness and cognition. Please include it as covariate.

2. Age categorization (18–64 vs. ≥65) should be justified, as this choice may obscure heterogeneity within older age groups.

Results/Discussion

1. Use of “chronic loneliness” or “persistent loneliness” with cross-sectional measurement:

The Results section refers to “chronic loneliness or persistent loneliness,” but loneliness appears to be measured at a single time point without information on duration or persistence. Without longitudinal or retrospective data on how long loneliness has been experienced, it is unclear how chronicity is being operationalized. The authors should clarify what they mean by “chronic” in this context or revise the terminology to avoid implying temporal persistence that is not measured.

2. Ambiguous terminology (e.g., “frequently married”):

The phrase “frequently married” is unclear and potentially misleading.

3. Overinterpretation of baseline group differences in a very large sample:

The statement that baseline characteristics “highlight pronounced socioeconomic, psychosocial, and demographic differences” may overstate the substantive meaning of these differences. Given the very large sample size, statistically significant differences are expected even when effect sizes are small. The authors should be cautious in interpreting these differences and consider reporting or referencing effect sizes, or rephrasing to avoid implying meaningful group separation based solely on statistical significance.

4. Dose-Response Interpretation:

While the graded pattern across loneliness categories is visually compelling, the interpretation of a strong “dose–response relationship” warrants additional verification. Given the very large sample size, statistically significant differences across all categories may reflect high power rather than substantively meaningful contrasts. I encourage the authors to supplement pairwise comparisons with additional robustness checks, such as reporting effect sizes (e.g., absolute risk differences), conducting a formal test for trend, or evaluating potential non-linear or threshold effects (e.g., contrasting high-frequency loneliness vs. lower-frequency groups). This would strengthen confidence that the observed pattern reflects a meaningful dose–response relationship rather than precision alone.

Also, the dose–response framing implicitly assumes a monotonic relationship in which increasing frequency of loneliness is uniformly more harmful. However, prior loneliness research suggests that occasional or moderate loneliness may be normative and not necessarily detrimental, whereas adverse effects may emerge only beyond certain thresholds of frequency or severity. The authors may wish to consider alternative characterizations of the pattern (e.g., threshold or non-linear effects) or to discuss why a linear or monotonic interpretation is theoretically justified in this context.

Clarifying these issues would strengthen the interpretation of the findings and avoid overstatement of dose–response effects based solely on categorical contrasts in a cross-sectional, high-powered dataset.

5. Causal and temporal language not supported by the data:

Phrases such as “detrimental impact,” “consequences,” and “heightened vulnerability” imply a causal or longitudinal interpretation. Given the cross-sectional design and single-time measurement of loneliness and subjective cognitive decline, the authors should avoid language suggesting effects or consequences and instead describe these findings as differences in associations or predicted probabilities.

6. Outcome Interpretation and Terminology:

The results and discussion section incorrectly refers to the outcome as “predicted cognitive function”. However, the outcome is subjective cognitive decline, operationalized as a binary self-report of worsening confusion or memory over the past 12 months. This measure does not capture cognitive function or performance per se, but rather perceived cognitive change. Even though SCD has been associated with objective cognitive impairment and dementia risk in prior studies, describing results as differences in “cognitive function” overstates what is measured. The authors should revise the language throughout to refer to predicted probability of subjective cognitive decline or self-reported cognitive difficulties, rather than cognitive function.

7. Interpretation of Sex Differences in High Loneliness:

The explanation offered for the observed sex difference among individuals reporting persistent loneliness is not well aligned with the results or the constructs measured in this study. While the authors report that women who are always lonely have higher predicted probabilities of subjective cognitive decline than men, the proposed explanation—that women’s family and social networks have weakened—invokes objective social network characteristics that were neither measured nor analyzed. This is particularly problematic given the authors’ earlier emphasis that loneliness and social isolation are related but distinct constructs.

As currently written, this explanation appears speculative and not directly supported by the data. The authors should either provide empirical evidence linking sex differences in loneliness-related cognitive complaints to network disruption, or reframe this interpretation more cautiously (e.g., in terms of differential psychological, emotional, or perceptual responses to loneliness among women). Clarifying this distinction would improve conceptual consistency and strengthen the discussion.

8. Racial/Ethnic Differences and Cultural Interpretation:

The discussion of racial/ethnic differences raises an important point about potential cultural variation in how loneliness relates to cognition, but this section would benefit from further development. Statements referring to “stronger family and community networks,” “culturally distinct interpretations of loneliness,” and “differences in reporting SCD” are plausible but remain speculative and are not sufficiently elaborated.

Given that the authors invoke “complex cultural differences” to interpret these findings, it would strengthen the discussion to expand on what these differences may entail, drawing more explicitly on prior literature (e.g., cultural norms around emotional expression, familism, stigma, or differential meaning of loneliness and cognitive complaints across racial/ethnic groups).

6. PLOS authors have the option to publish the peer review history of their article (what does this mean?). If published, this will include your full peer review and any attached files.

Do you want your identity to be public for this peer review? For information about this choice, including consent withdrawal, please see our Privacy Policy.

Reviewer #1: No

Reviewer #2: No

---

## [Author Response · Author response to Decision Letter 1]

19 Mar 2026

PONE-D-25-65508

Manuscript Title: Loneliness and Cognitive Decline Among U.S. Adults: A Stratified Analysis of the BRFSS

Journal: PLOS ONE

Dear Academic Editor and Reviewers,

We thank the Editor and Reviewers for their careful evaluation of our manuscript and for their thoughtful and constructive comments . We have carefully revised the manuscript to address all concerns raised, including clarifications to the methodology, refinement of terminology, strengthening of the analytical approach, and revision of the interpretation of findings to ensure appropriate caution and clarity. All changes have been incorporated into the revised manuscript and are clearly indicated. Detailed, point-by-point responses to each reviewer comment are provided below.

Reviewer #1: Thank you for the opportunity to read this interesting and important paper that investigated the relationship between the frequency of loneliness and subjective cognitive decline. The authors found that the frequency of loneliness was associated with subjective cognitive decline, showing predicted probability of SCD of 9.9% in respondents that never felt lonely, all the way up to 45% in those that always felt lonely. The paper is well written, clearly communicates study methods, and adds valuable insights into the broader literature on impacts and significance of social disconnection and demonstrates a thoughtful analysis to support whether interventions could be worthwhile. There are a few comments that the authors could address to strengthen the piece.

Major comments:

1.) The language used in certain parts of the paper needs adjustment.

- The study conclusions, "These findings identify loneliness as a modifiable social determinant of cognitive health" could be more cautiously and grounded in study findings

- The claim that a one-time cross-sectional assessment of loneliness describes chronicity of loneliness is a big claim, I believe the authors are limited since BRFSS doesn't include longitudinal data that would be more appropriate to look at trends for individuals rather than 1 time assessment of "chronicity". The primary difference between frequency and chronicity is in their focus: frequency measures how often an event occurs within a specific period, while chronicity measures how long a condition persists over a long-term, sustained period, there is a slight difference given what data is available and the wording should reflect this.

-There are several instances through the manuscript that outcome of "cognitive decline" is referred to without listing "subjective", it is important to include subjective throughout to minimize confusion, consider including in title and tables as well as text.

RESPONSE

Thank you for your thoughtful review of our manuscript. We appreciate your constructive feedback and have revised the paper accordingly.

Language and conclusions: We agree that our original conclusions were too strong given the cross‑sectional design. We have revised the abstract and discussion to emphasize that our study identifies associations between loneliness and subjective cognitive decline rather than implying causality. The conclusions now refer to loneliness as a potentially modifiable social factor and note that longitudinal studies are needed to determine causal effects (Lines 447-455, Pg. 21-22).

Frequency vs. chronicity of loneliness: We acknowledge that the BRFSS question measures how often respondents feel lonely at a single time point and does not capture the duration or chronicity of loneliness. Throughout the manuscript we replaced terms like “chronic loneliness” with “frequent loneliness (i.e., always or usually feeling lonely).” We also added a sentence in the Limitations section highlighting that our measure assesses perceived frequency rather than persistent loneliness and that we cannot infer chronicity from these data (Lines 385-396, Pg. 19).

Use of “subjective” in cognitive decline: To avoid confusion, we have ensured that the outcome is consistently referred to as “subjective cognitive decline” (SCD). We updated the title, short title, tables and text to include “subjective” wherever appropriate.

We hope that these revisions address your concerns and improve the clarity and accuracy of the manuscript. We appreciate your helpful suggestions and hope the updated version meets the standards for publication.

2.) Respondents that were excluded from the analyses because of "don't know, refused or missing key variable"- team may want to describe who is missing from their analysis. It is also possible that individuals with greatest loneliness frequency do not make it into a study like BRFSS and this limitation could be acknowledged.

RESPONSE:

Thank you for this thoughtful comment. We have addressed it by (1) clarifying in the Study population section that respondents were excluded if they answered “don’t know/not sure,” refused, or had missing data for loneliness, SCD, or covariates, and that excluded individuals tended to be older, less educated, and more likely from racial/ethnic minority groups; (2) adding a limitation noting that exclusion of these respondents may bias results if missingness is related to loneliness or cognitive decline; and (3) adding another limitation acknowledging that telephone-based surveys such as the BRFSS may under-sample highly lonely individuals who lack phone service or are socially isolated, citing evidence that nonresponse and coverage limitations can materially affect survey estimates. We believe these changes improve transparency about sample composition and the potential impact of nonresponse on our findings (Lines 414-422, Pg. 20).

3.) The authors clearly communicate study methods and it makes sense the use of adjusted marginal probabilities, but unclear if best way to assess dose response would be pairwise comparisons rather than trend test, and if pairwise is used then would consider the need for multiple pairwise comparisons correction to control the likelihood of false positives.

RESPONSE

Thank you for this insightful comment. In our study, the loneliness categories (“never,” “rarely,” “sometimes,” “usually,” “always”) are naturally ordered, and our goal was to test whether the prevalence of subjective cognitive decline (SCD) increases in a monotonic, dose‑response fashion across these categories. We therefore used a test for trend within a logistic‑regression framework rather than conducting pairwise comparisons between each category. Trend tests are appropriate when the exposure is ordinal and provide a single hypothesis test without requiring adjustment for multiple comparisons. By contrast, pairwise comparisons among five categories would involve numerous tests and, as highlighted in the statistical literature, would require multiplicity corrections (e.g., Bonferroni or Dunnett adjustments) to control the Type I error rate. We have added language in the Methods section explaining our use of the trend test and, in the Results, we now note that supplementary pairwise comparisons with Bonferroni correction yielded similar findings

4.) The authors appropriately completed sensitivity analyses, but do not state which auxillary variables were included in their imputation models and this should be clarified as well as justification that missing at random assumption was met.

RESPONSE

Thank you for pointing this out. We have revised the Methods section to specify the auxiliary variables included in our multiple‑imputation models. In addition to age, sex, race/ethnicity, education, and income (used in the primary analyses), we now state that the imputation model also included marital status, employment status, general health, smoking status, physical activity, and household size as auxiliary predictors of missingness. We further note that the missing‑at‑random (MAR) assumption is plausible because missingness was associated with these observed characteristics; by including predictors of non‑response and of the missing values themselves, the imputation model helps satisfy the MAR assumption (Lines 232-238, Pg. 11). We have also added a statement in the Limitations acknowledging that the MAR assumption cannot be fully tested (Lines 423-425, Pg. 20).

5.) One area that requires more consideration and explanation, Table 1 presents total population 86K and proportion across loneliness categories, only 2.4% of population always lonely and only 2.71% usually lonely (while Never lonely is ~45% and sometimes ~21%, and rarely ~30%). Given small proportion of individuals in these categories there is concern when using these group to calculate average marginal effects that there could be significant errors in estimation due to high variance and small sample size bias. While the marginal effects themselves might not be biased, the precision of these estimates (the standard errors) is often severely compromised, leading to unreliable inferences.

RESPONSE

Thank you for raising this important point. We appreciate this comment. Although the proportion of respondents reporting “always lonely” (2.4%) and “usually lonely” (2.7%) appears small, the large overall sample size (N ≈ 86,000) results in more than 2,000 individuals in each of these categories. Logistic regression and marginal effects estimation rely on the full sample rather than subgroup-specific models, and therefore the estimation remains statistically stable. In addition, the number of observations and outcome events in these categories substantially exceeds commonly cited thresholds for reliable logistic regression estimation. Precision of estimates is reflected in the reported standard errors and confidence intervals, which remain narrow across categories, suggesting that the estimates are not unduly affected by small-sample variability. We have added a sentence to the Statistical Analysis section clarifying that the “always” and “usually” categories contained more than 2 000 respondents each, that we used robust variance estimation, and that sensitivity analyses confirmed that results were not driven by small cell counts (Lines 193-202, Pg.9). We have also noted in the Discussion that estimates for the smallest categories should be interpreted with appropriate caution (Lines 426-430, Pg. 21).

6.) The authors previous paper explores associations between loneliness and mental health (Akinyemi O, Abdulrazaq W, Fasokun M, Ogunyankin F, Ikugbayigbe S, Nwosu U, Michael M, Hughes K, Ogundare T. The impact of loneliness on depression, mental health, and physical well-being. PLoS One. 2025 Jul 9;20(7):e0319311. doi: 10.1371/journal.pone.0319311. PMID: 40632698; PMCID: PMC12240311.), which is likely relevant to subjective cognitive decline, but this is not examined in these analyses, unclear why. The paper states that they" lacked information on potentially confounding factors such as depressive symptoms", although the BRFSS has these measures, this needs to be clarified.

RESPONSE:

Thank you for your careful reading and for pointing this out. We acknowledge that our prior BRFSS‑based study focused on depression and mental‑health days, whereas the present manuscript specifically addresses subjective cognitive decline. We have revised the introduction to reference our earlier work and to explain that depression and mental‑health problems are known risk factors for cognitive decline and dementia (Lines 108-112, Pg. 5). We have also clarified in the Methods section that while the BRFSS includes questions on poor mental‑health days and lifetime diagnosis of depressive disorder, these variables were not included as covariates in our primary models because (a) they may lie on the causal pathway between loneliness and cognitive decline (adjusting for them could attenuate the total effect of loneliness), and (b) the depressive‑disorder item reflects lifetime diagnosis rather than current depressive symptoms and was not consistently reported across all states and years. In the Discussion we note this decision as a limitation and encourage future work to explore the role of mental‑health variables as potential mediators.

A few additional minor comments:

1.) In the methods section of abstract age range listed is "16-66" but minimum age of those in study was 18, this should be clarified or fixed.

RESPONSE

Thank you for pointing this out. Our analytic sample included respondents aged 18–66 years, not 16–66. The “16” in the abstract was a typographical error. We have corrected the abstract to clarify that the analysis includes only adults 18 years or older (Lines 27-34, Pg.2).

2.) Limitation of cross-sectional studies listed in introduction seems erroneous as the current study is also cross sectional, authors may want to be reword/ get more specific that other studies have looked at loneliness as binary yes/no without considering frequency/dose which is what it seems they are trying to convey.

RESPONSE

Thank you for pointing out this inconsistency. We have revised the Introduction to clarify that we were critiquing prior cross‑sectional studies for dichotomizing loneliness (yes/no) and thus failing to capture the frequency or “dose” of loneliness. Our study is cross‑sectional as well, so we no longer refer to cross‑sectional design per se as a limitation in the introduction. Instead, we now note that previous analyses often used a binary loneliness measure, whereas our work extends this by examining loneliness frequency categories (Lines 76-88,Pg. 4). The Discussion continues to acknowledge that our cross‑sectional design precludes causal inference.

3.) The BRFSS study question on how often participants feel lonely does not include a timeframe, like over the past X amount of time, so it's unclear how participants may interpret this question, while question regarding SCD has clear time delineation of 12 months. Most other BRFSS study questions consider 30 day timeframe so its possible that is assumed by the participants, this could be conveyed more clearly for readers that are not familiar with BRFSS

RESPONSE

Thank you for raising this point. We agree that the BRFSS loneliness question (“How often do you feel lonely? Is it … always, usually, sometimes, rarely or never?”) does not specify a reference period. In contrast, the subjective cognitive decline (SCD) question explicitly asks whether confusion or memory loss has been happening more often or getting worse “during the past 12 months.” To clarify this for readers unfamiliar with BRFSS, we have revised the Method section to note explicitly that the loneliness question lacks a defined timeframe and therefore captures respondents’ general perception of their loneliness rather than loneliness experienced within a specific period. We also point out that many BRFSS questions refer to the past 30 days, which could lead some respondents to anchor their answers to that interval, but the absence of a timeframe introduces variability in interpretation (Lines 148-154, Pg. 7). This difference in recall period is now discussed as a limitation in the Discussion (Lines 386-392, Pg.19).

4.) Methods section shares covariates were selected 'based on prior research', authors should include reference for what is being referred to.

RESPONSE

Thank you for bringing these points to our attention. We have revised the Methods section to cite a narrative review on loneliness and cognitive decline that recommends controlling for covariates across five domains—socio‑demographic, social health, health behaviours, physical health, and mental health. This citation clarifies the evidence base that guided our choice of covariates. In the Discussion, we now acknowledge three recent papers: (1) Kang et al. (2025), which reports that chronically lonely adults show less improvement in working memory and processing speed over two years; (2) Luchetti et al. (2025), which finds that both between‑ and within‑person fluctuations in loneliness are associated with daily subjective cognitive concerns; and (3) Ren et al. (2025), which shows that prolonged loneliness increases the risk of incident cognitive decline and dementia by about 31 %. These studies strengthen the growing evidence base linking loneliness to cognitive health and support the rationale for our work

5.) Some additional more recent references to consider that could be used to situate thi

---

## [Decision Letter · Decision Letter 1]

21 Apr 2026

Loneliness and Cognitive Decline Among U.S. Adults: A Stratified Analysis of the BRFSS

PONE-D-25-65508R1

Dear Dr. Fasokun,

We’re pleased to inform you that your manuscript has been judged scientifically suitable for publication and will be formally accepted for publication once it meets all outstanding technical requirements.

Kind regards,

Alessia Tessari, Ph.D.

Academic Editor

PLOS One

Additional Editor Comments (optional):

Dear Authors,

I am pleased to inform you that, following the completion of the review process, your manuscript entitled “Loneliness and Cognitive Decline Among U.S. Adults: A Stratified Analysis of the BRFSS” has been accepted for publication in PLOS One.

Your revisions have satisfactorily addressed the comments raised during the review process.

Your manuscript will now proceed to the production stage. You will receive further communication regarding proofs and publication details in due course.

Thank you for choosing PLOS One for the dissemination of your work.

Kind regards,

Alessia Tessaru

Reviewers' comments:

Reviewer's Responses to Questions

Comments to the Author

1. If the authors have adequately addressed your comments raised in a previous round of review and you feel that this manuscript is now acceptable for publication, you may indicate that here to bypass the “Comments to the Author” section, enter your conflict of interest statement in the “Confidential to Editor” section, and submit your "Accept" recommendation.

Reviewer #1: All comments have been addressed

Reviewer #2: All comments have been addressed

2. Is the manuscript technically sound, and do the data support the conclusions?

Reviewer #1: Yes

Reviewer #2: Yes

3. Has the statistical analysis been performed appropriately and rigorously? 

Reviewer #1: I Don't Know

Reviewer #2: Yes

4. Have the authors made all data underlying the findings in their manuscript fully available?

Reviewer #1: Yes

Reviewer #2: Yes

5. Is the manuscript presented in an intelligible fashion and written in standard English?

Reviewer #1: Yes

Reviewer #2: Yes

6. Review Comments to the Author

Reviewer #1: The authors were fully responsive to suggested edits which have strengthened the paper.

No additional recommendations

Reviewer #2: (No Response)

7. PLOS authors have the option to publish the peer review history of their article (what does this mean?). If published, this will include your full peer review and any attached files.

Do you want your identity to be public for this peer review? For information about this choice, including consent withdrawal, please see our Privacy Policy.

Reviewer #1: No

Reviewer #2: No

---

## [Editor Report · Acceptance letter]

PONE-D-25-65508R1

PLOS One

Dear Dr. Fasokun,

I'm pleased to inform you that your manuscript has been deemed suitable for publication in PLOS One. Congratulations! Your manuscript is now being handed over to our production team.

Kind regards,

on behalf of

Professor Alessia Tessari

Academic Editor

PLOS One